# Targeted Therapy in Follicular Lymphoma: Towards a Chemotherapy-Free Approach

**DOI:** 10.3390/cancers15184483

**Published:** 2023-09-08

**Authors:** Chung-Jiah J. Chen, Michael Y. Choi, Benjamin M. Heyman

**Affiliations:** 1Department of Medicine, Division of Hematology-Oncology, UC San Diego Health, La Jolla, CA 92037, USA; cjc052@health.ucsd.edu (C.-J.J.C.); mychoi@health.ucsd.edu (M.Y.C.); 2Department of Medicine, Division of Regenerative Medicine, UC San Diego Health, La Jolla, CA 92037, USA

**Keywords:** follicular lymphoma, targeted therapy, molecular testing

## Abstract

**Simple Summary:**

Follicular lymphoma (FL)—a common, indolent, and usually non-curable B-cell non-Hodgkin lymphoma—is frequently treated with cytotoxic chemotherapy with anti-CD20 therapy. The downsides of this approach include cumulative toxicities that limit the total duration of treatment and may be prohibitive in unfit patients, as well as chemotherapy-refractory disease that develops over time. Targeted therapies in FL show promise by inhibiting key molecular pathways responsible for cancer proliferation and survival, reducing toxicity, and potentially circumventing chemotherapy refractoriness. In this review, we investigate, summarize, and critique the sentinel studies underpinning the use of targeted therapies currently available for FL in both the front-line and relapsed–refractory setting. We also detail the role of molecular markers in FL as well as future directions in targeted treatment for follicular lymphoma.

**Abstract:**

Background: The treatment of follicular lymphoma (FL) has previously centered on chemoimmunotherapy, which can be disadvantageous due to patient intolerance, cumulative toxicities, and disease refractoriness. Targeted therapies can produce deep responses and improve progression-free and overall survival with more tolerable adverse event profiles. Methods: We summarize the current literature and key clinical trials regarding targeted therapies in follicular lymphoma both in the front-line and in the relapsed-refractory setting. Results: Targeted therapies studied in FL include immune modulators, anti-CD20 antibodies, Bruton’s tyrosine kinase (BTK) inhibitors, enhancers of zeste homolog 2 (EZH2) inhibitors, phosphoinositide 3-kinase (PI3K) inhibitors, and B-cell lymphoma 2 (BCL-2) inhibitors. Chimeric antigen receptor (CAR-T) therapy and bispecific T-cell engager (BiTE) therapies also show promise in monotherapy and in combination with targeted therapies. These therapies exhibit high overall response rates and substantial progression-free survival and overall survival, even in high-risk patients or patients previously refractory to chemotherapy or rituximab. Adverse events vary substantially but are generally manageable and compare favorably to the cumulative toxicities of chemotherapy. Conclusion: Targeted therapies represent a paradigm shift in the treatment of FL. Further studies are needed to directly compare these targeted therapies and their combinations, as well as to investigate biomarkers predictive of response.

## 1. Introduction

Follicular lymphoma (FL) is the most common indolent B-cell non-Hodgkin lymphoma (NHL) in the United States [1]. Although the typical clinical presentation of FL is indolent, with some patients requiring treatment only after meeting certain criteria such as the Groupe d’Etude des Lymphomes Folliculaires (GELF) criteria, some patients will experience transformation into an aggressive lymphoma phenotype that necessitates more expedient treatment. Initial treatment regimens with chemotherapy paired with anti-CD20 therapy, although non-curative, have been effective in extending life expectancy to more than 10 years in many cases [2]. However, patients who are unfit for chemotherapy, who have cumulative chemotherapy-related toxicities from prior treatment, who have early disease progression, or who are refractory to initial chemotherapy all would benefit from non-chemotherapy-based treatment. Several targeted therapies have been developed and investigated for FL. These targeted therapies offer distinct adverse-effect (AE) profiles from traditional cytotoxic chemotherapy and provide therapeutic options for those with chemotherapy-refractory disease.

In this review, we briefly discuss the mechanisms of actions of targeted therapies in follicular lymphoma. We searched the PubMed archive of the National Library of Medicine of the United States to identify the most relevant and sentinel studies for each targeted therapy. We discuss the various regimens that have been studied in the relapsed–refractory setting as well as the front-line setting. We also briefly discuss novel immune- and cellular-based therapies, bispecific T-cell engager therapies, and chimeric antigen receptor T-cell therapy and how these therapies may be used in combination with targeted therapy. We also discuss the current state of molecular testing in FL, its current limitations, and how it may be used to guide therapy selection in the future.

## 2. Overview of Targeted Therapies

Several targeted therapies have been studied in follicular lymphoma. Many of these agents have been previously studied in other hematologic malignancies and are being adapted for use in FL. A visual overview of the mechanisms of these therapies is summarized in Figure 1.

Lenalidomide is an immune-modulatory agent widely used in B-cell NHLs, such as multiple myeloma and mantle cell lymphoma [3]. It binds the cereblon E3 ubiquitin ligase complex, which results in the increased ubiquitination and degradation of the transcription factors Aiolos and Ikaros in T lymphocytes [3,4]. This leads to a host of downstream effects, including a release of cytokines that upregulates cytotoxic T cells and natural killer cells, which results in the enhancement of immune response in the tumor microenvironment. Through cytokine regulation, lenalidomide also induces cell cycle arrest, decreases the expression of anti-apoptotic proteins, and inhibits angiogenesis and other stromal support mechanisms [4].

Rituximab, a type I chimeric anti-CD20 monoclonal antibody, has long been a mainstay of therapy for B-cell lymphomas, and in FL, it is commonly used in front-line, subsequent-line, and maintenance therapies in combination with chemotherapy or as monotherapy [5,6]. Efforts to enhance anti-CD20 activity have resulted in obinutuzumab, a type II glycoengineered anti-CD20 monoclonal antibody that has been shown in previous in vitro and in vivo studies to enhance B-cell depleting activity and increase antibody-dependent cellular phagocytosis when compared to rituximab [5]. In rituximab-refractory FL patients, obinutuzumab with bendamustine has been shown to be superior to obinutuzumab alone, suggesting that its enhanced anti-CD20 activity when compared to rituximab can be translated to meaningful clinical benefit even in patients who were pre-treated with rituximab [7]. Furthermore, in the GALLIUM study, FL patients who received obinutuzumab with chemotherapy followed by obinutuzumab maintenance demonstrated improved PFS compared with those patients who received rituximab with chemotherapy followed by rituximab maintenance, suggesting that obinutuzumab may be more active in the front-line setting than rituximab [8,9].

Covalent Bruton tyrosine kinase inhibitors (BTKi) disrupt signaling pathways downstream from the B-cell antigen receptor (BCR), therefore inhibiting key proliferation and maintenance pathways critical for the survival of B-cell malignancies [10]. Several BTKi, including ibrutinib, acalabrutinib, and zanubrutinib, are currently widely used in multiple indolent B-cell malignancies, including chronic lymphocytic leukemia (CLL), marginal zone lymphoma, and mantle cell lymphoma [11]. Zanubrutinib and acalabrutinib are both second-generation BTKi that are intended to be more selective than the first-generation ibrutinib, which may result in an overall decreased incidence of adverse events, although direct comparisons are lacking currently [12]. A new class of non-covalent BTKi (which includes pirtobrutinib) is intended to sidestep acquired BTK binding-site mutations that confer resistance to the first- and second-generation BTK inhibitors. Pirtobrutinib has been recently approved for use by the U.S. Food and Drug Administration (FDA) in mantle cell lymphoma [12]. Currently, ibrutinib and zanubrutinib have been studied in FL [11,13,14,15].

Venetoclax is a selective B-cell lymphoma 2 (BCL-2) inhibitor currently used as monotherapy or in combination with other agents in a wide range of hematologic malignancies, including CLL, myelodysplastic syndrome, and acute myeloid leukemia [16,17,18]. This inhibition of pro-survival BCL-2 results in mitochondrial-mediated apoptosis in BCL-2-dependent hematologic malignancies [19].

Tazemetostat is a first-in-class, oral small molecule enhancer of zeste homolog 2 (EZH2) inhibitor that has shown promise in treating NHLs [20]. EZH2 is an important epigenetic regulator of normal germinal center formation in B cells, and approximately 20% of patients with follicular lymphoma have gain-of-function mutations in EZH2. These pathogenic activating EZH2 mutations lead to epigenetic silencing and ultimately block terminal differentiation, resulting in the accumulation of malignant clones [20]. Tazemetostat blocks EHZ2 activity, thus selectively inhibiting the ability of FL cells to proliferate. Notably, all FL cells are dependent on EZH2 to some extent, regardless of EZH2 mutational status [21].

Copanlisib is the sole remaining phosphoinositide 3-kinase (PI3K) inhibitor that remains approved for the treatment of FL by the FDA. Aberrant PI3K pathway signaling is a frequently activated pathway in malignancy, and multiple PI3K inhibitors have previously been approved for a wide range of solid and hematologic malignancies [22]. Previously, idelalisib, duvelisib, and umbralisib were all FDA-approved for FL based on single-arm trial data showing favorable response rates. However, subsequent randomized control trials have raised concerns about serious adverse events, including immune-mediated toxicity and serious infections, resulting in decreased overall survival in some treatment arms [23]. Idelalisib and duvelisib have been voluntarily withdrawn for FL treatment, and umbralisib remains under investigation for these concerns [23]. Additional PI3K inhibitors that were under development include parsaclisib and zandelisib. Parsaclisib demonstrated a high overall response rate (ORR) of 75.4% and median progression-free survival (PFS) of 14.0 months—however, the New Drug Application (NDA) for parsaclisib was withdrawn in 2022 as the FDA requested confirmatory studies that its manufacturer elected not to pursue [24,25]. Similarly, zandelisib monotherapy was associated with high ORR (72.5%) with an estimated median PFS of 11.4 months—however, further development of zandelisib outside of Japan was stopped abruptly in 2022 after the FDA requested further changes to the phase 3 COASTAL trial [26,27].

Bispecific T-cell-engaging (BiTE) antibodies stimulate directed cytotoxic T-cell responses against oncologic targets and have been employed in treating B-cell malignancies, including follicular lymphoma [28,29,30]. BiTE antibodies simultaneously bind to CD3 receptors on effector T cells and a specific target receptor on cancer cells, facilitating T-cell engagement towards the targeted cancer cells, synapse formation, and ultimately resulting in antibody-dependent cellular toxicity (Figure 2A) [29,30]. For the BiTE antibodies investigated for use in FL, the targeted receptor on cancer cells is CD20. Mosunetuzumab is an IgG1-based, bispecific CD20-directed CD3 T-cell engager that has received accelerated approval by the FDA for relapsed–refractory FL [28,31]. Epcoritamab is an IgG1 CD3xCD20 BiTE antibody that can be administered subcutaneously (compared to intravenous administration for other BiTE options) and has demonstrated activity in DLBCL, FL, and MCL [32]. Glofitamab is a CD3xCD20 BiTE antibody that has a somewhat different structure: it has two Fab regions for CD20 (derived from the higher-affinity binding regions of obinutuzumab) and one Fab region for CD3, which results in higher binding avidity to the target B-cell and hypothesized greater resulting potency [33]. Odronextamab is a first-in-class, hinge-stabilized, fully human IgG4-based CD3xCD20 BiTE antibody that has been preliminarily studied in B-cell non-Hodgkin lymphomas, including follicular lymphoma [34,35,36]. Studies combining BiTE therapy with other targeted agents, such as lenalidomide, are underway [37,38]. Compared to CAR-T therapy, BiTE therapy represents an “off-the-shelf” T-cell-mediated antitumor therapy, bypassing the logistical difficulties of T-cell collection and lead time for product manufacturing.

Anti-CD19-directed chimeric antigen receptor T-cell (CAR-T) therapy represents a major advancement in cellular and immunotherapy against B-cell malignancies such as DLBCL, mantle cell lymphoma, and follicular lymphoma [39,40]. These cellular therapies are engineered autologous T-cell products that center on a chimeric antigen receptor (CAR). The CAR construct contains an extracellular target-binding domain (anti-CD19), an intracellular essential activating domain (CD3 ζ), and a costimulatory signaling domain (CD28 in axi-cel and 4-1-BB in tisa-cel) (Figure 2B) [39,41]. CAR-T therapy requires harvesting the patient’s autologous T-cells, transfecting the cells with a viral vector that contains the CAR construct, expanding the CAR-T cells, and then re-infusing the CAR-T therapy into the patient, who previously received lymphodepleting therapy. The anti-CD-19 binding domains of the CAR-T cells then bind CD-19 on the tumor cells, inducing CAR-T proliferation and cytotoxic responses that destroy the tumor cell [40]. Currently, axicabtagene ciloleucel (axi-cel) and tisagenlecleucel (tisa-cel) have been studied in FL and approved by the U.S. FDA [42,43]. Additionally, preliminary results for lisocabtagene maraleucel (liso-cel) are available [44]. Overall, CAR-T therapies have shown high response rates and durable responses but also have a unique adverse effect profile, including cytokine release syndrome (CRS) and immune-effector-cell-associated neurotoxicity syndrome (ICANS) [45]. Additionally, logistical difficulties, including apheresis procedures for T-cell collection and extended requisite lead-time for manufacturing of the product, may prove limiting for patients who require a response in a short time frame [46].

## 3. Therapies in the Relapsed/Refractory (R/R) Setting

Although most patients with follicular lymphoma generally respond well to initial chemotherapy with anti-CD20 therapy in the front-line setting, options for subsequent-line treatment were previously more limited to a re-challenge of chemotherapy agents based on duration of response to prior therapies, fitness for therapy, and cumulative toxicities from prior chemotherapy treatment. Furthermore, overall response duration tends to decrease with each subsequent line of chemoimmunotherapy. A retrospective analysis of 569 FL patients between 2001 and 2014 treated with combination rituximab with chemotherapy showed that median PFS decreased from 10.6 years after first-line therapy to 2.4 years after second-line therapy to 2 years after third-line therapy [47]. This decrease in PFS for subsequent-line treatment highlights the urgent need for novel targeted therapies that demonstrate good response in the subsequent line.

The advent of targeted therapies in the relapsed/refractory setting has provided a plethora of new, non-chemotherapy treatments that show response even in patients with poor prognostic features, such as progression of disease within 24 months of initial treatment (POD24) or high-risk Follicular Lymphoma International Prognostic Index (FLIPI) scores on diagnosis. Table 1 summarizes the relevant studies supporting the use of targeted therapies in follicular lymphoma in the relapsed–refractory setting.

### 3.1. Rituximab with Lenalidomide (R^2^) in the Relapsed/Refractory Setting

The efficacy of R^2^ in relapsed/refractory follicular lymphoma was studied in the randomized, phase III, multicenter AUGMENT trial [48], which randomized 358 patients with either follicular lymphoma (FL) or marginal-zone lymphoma to two study arms: one in which the patients received R^2^ and the other in which the patient received rituximab with placebo. The FL patients had Grade 1–3a disease, an advanced-stage disease requiring treatment, and had received at least one prior line of therapy. The patients enrolled in the R^2^ arm demonstrated significantly improved progression-free survival of 39.4 months vs. 14.1 months in the placebo arm [48]. Although AUGMENT was not powered to detect overall survival, five-year overall survival rates were significantly higher in the R^2^ arm (83.2%) versus 77.3% in the rituximab/placebo group [48,49].

Overall, R^2^ was found to be tolerable with an adverse effect profile that was manageable. As expected, patients in the R^2^ arm experienced more adverse events (AE) than those who received rituximab monotherapy, including higher rates of Grade 3 or 4 neutropenia and leukopenia that necessitated growth factors in 36% of the R^2^ group [48]. AEs classically associated with lenalidomide, including venous thromboembolism, cutaneous reactions, and diarrhea, were also higher in the R^2^ group. However, the AE profile of R^2^ was generally considered to be tolerable and manageable [49].

A major limitation of the AUGMENT trial is that it does not directly compare R^2^ to other subsequent-line chemotherapy regimens, such as bendamustine-rituximab or bendamustine-obinutuzumab [7,62]. In the GADOLIN study, which evaluated bendamustine-obinutuzumab in the relapsed/refractory setting, PFS and OS appear to be numerically similar to R^2^ in AUGMENT, and both R^2^ and bendamustine-obinutuzumab demonstrated high rates of neutropenia [7,48]. However, AUGMENT demonstrated PFS benefit in those patients with prior chemotherapy resistance, prior rituximab resistance, or more than one prior treatment regimen. Therefore, the most important utility of R^2^ may be as a potential therapy option for those with chemotherapy-resistant or heavily pre-treated disease.

### 3.2. Obinutuzumab with Lenalidomide in the Relapsed/Refractory Setting

Early data from a phase I/II trial of obinutuzumab and lenalidomide in indolent, relapsed NHL patients (of which 57/66 patients had Grade 1–3a FL) showed a promising ORR of 98%, CR rate of 72%, with a 24-month PFS of 73% [51]. This regimen was overall well-tolerated, with thrombocytopenia, neutropenia, fatigue, rash, and cough being the most prominent Grade 3 or 4 adverse events [51].

The GALEN trial was designed to better evaluate obinutuzumab and lenalidomide in relapsed/refractory FL. This single-arm, multicenter phase Ib/2 study of obinutuzumab plus lenalidomide enrolled 86 patients with CD-20 positive Grade 1–Grade 3a FL who previously received at least one prior rituximab-containing regimen.

In the GALEN study, the overall response rate of 79% and complete response rate of 38% is numerically superior to the R^2^ regimen in the AUGMENT trial (78% ORR, 34% CR) [48,50]. The two-year PFS rate was 65%, and the two-year OS rate was 87%, also comparable numerically to R^2^ in AUGMENT [48,50]. The most notable Grade 3–4 adverse events were neutropenia (44%) and thrombocytopenia (14%). Notably, 16% of patients also had infusion-related reactions, 15% had serious infections, and 9% of patients had second primary malignancies, including cutaneous malignancies and myelodysplastic syndrome.

It is currently less clear if the PFS results favoring obinutuzumab-based chemoimmunotherapy over rituximab-based chemoimmunotherapy in the previous GALLIUM study translate to the lenalidomide-containing analogs, GALEN vs. R^2^ [8]. Although the GALEN study did not directly compare the GALEN regimen to the R^2^ regimen, the response rate is numerically superior to the response rate of R^2^ in other studies, such as AUGMENT, and the adverse event profile appears similar to R^2^ [50]. Overall, the favorable response rate and side effect profile suggest that the GALEN regimen is a reasonable subsequent-line choice, even after patients fail a previous rituximab-containing regimen, and further studies directly comparing GALEN to R^2^ are needed to determine whether the theoretically superior anti-CD20 activity of obinutuzumab over rituximab translates to improved clinical outcomes when combined with lenalidomide.

### 3.3. Venetoclax and Rituximab with or without Chemotherapy in the Relapsed/Refractory Setting

The phase II open-label CONTRALTO study assigned patients to three treatment arms: venetoclax with rituximab (VEN + R), venetoclax with rituximab and bendamustine (VEN + BR), and rituximab with bendamustine (BR). Overall, patients in the VEN + R arm did significantly worse than in the other two arms, with ORR of 35% when compared to 84% in both the VEN + BR arm and the BR arm [52]. PFS at 18 months was also significantly worse for the VEN + R arm than both the VEN + BR arm and the BR arm [52]. Notably, patients in the VEN + BR arm had similar CR rates and response durability, but the addition of venetoclax resulted in higher overall toxicity, driven by hematologic (neutropenia) and gastrointestinal adverse events [52].

This difference in outcomes could be partially explained by the fact that patients were selected for the VEN + R arm at the investigator’s discretion—this resulted in the VEN + R arm having a higher percentage of patients with refractory disease or who were heavily pre-treated. Overall, the utility of venetoclax in combination with other targeted therapies or chemotherapy in the relapsed/refractory setting remains unclear, and further studies are required to evaluate optimal venetoclax dosing and optimal targeted therapy combinations.

### 3.4. Ibrutinib Monotherapy in the Relapsed/Refractory Setting

Ibrutinib monotherapy has been studied in a phase II open-label trial that enrolled 40 patients with FL who had previously progressed. Patients in this study demonstrated a 38% ORR with a 13% CR rate and a 2-year PFS rate of 20.4% with a median PFS of 14.0 months [11]. Patients who were rituximab-refractory or chemotherapy-refractory were noted to have lower ORR than those who were rituximab- and chemotherapy-sensitive, respectively. These relatively disappointing results may have been skewed by the relatively high percentage of previous-therapy refractory patients (35%) or the higher median number of prior therapies (3) than the patients in either the AUGMENT or GALEN studies [11,48,50].

Although this trial shows relatively lower efficacy for ibrutinib monotherapy, limitations in sample size, as well as a higher proportion of higher-risk patients, may have skewed the results. Further studies with ibrutinib or other newer-generation BTK inhibitors in combination with other targeted therapies may yield more promising results.

### 3.5. Zanubrutinib + Obinutuzumab in the Relapsed Refractory Setting

The combination of zanubrutinib with obinutuzumab in the relapsed–refractory setting was evaluated in the phase II ROSEWOOD trial, which expanded on early-phase Ib data showing efficacy with median PFS of 25 months in FL patients [63]. In the ROSEWOOD trial, 217 pts with relapsed/refractory FL and who had received two or more prior lines of therapy were randomized to zanubrutinib + obinutuzumab (ZO) or obinutuzumab monotherapy (O). The ORR of the ZO patients was significantly higher at 68.3% than the patients who received O (45.8%), with significantly improved median PFS of 27.4 months (ZO) vs. 11.2 months (O) [15]. The adverse effect profile of patients who received ZO was generally as expected for other BTKi therapy, with thrombocytopenia neutropenia, diarrhea, fatigue, and constipation being the most common [15].

Overall, the ZO combination shows improved ORR and CR rates as well as improved PFS when compared to obinutuzumab alone. When compared to BTKi monotherapy (ibrutinib), the ORR, CR, and PFS rates also appear to be numerically superior to ibrutinib monotherapy alone, although further study is needed to better elucidate these differences [11]. Overall, adverse events also appear to be similar to ibrutinib monotherapy. It is less clear how ZO compares to GALEN in the relapsed–refractory setting, although GALEN appears to have numerically superior ORR, CR, and PFS rates [50]. This notwithstanding, the ZO regimen is a tolerable, efficacious therapy choice for relapsed–refractory patients.

### 3.6. Tazemetostat in the Relapsed/Refractory Setting

Tazemetostat was evaluated in the relapsed/refractory setting in an open-label, single-arm phase 2 trial that enrolled 99 previously treated patients with FL, with 45% of patients having EZH2 mutations. Out of those patients with EZH2 mutations, the ORR was 69%, and 13% of patients achieved a complete response. Of the EZH2 wild-type patients, ORR was 35%, and 4% of patients achieved a complete response [21]. The median PFS was 13.8 months and 11.1 months in patients with EZH2 mutations and patients who were EZH2 wild-type, respectively. Overall, tazemetostat was relatively well-tolerated; common AEs included nausea, alopecia, diarrhea, asthenia, and fatigue. In total, 27% of patients experienced serious treatment-emergent adverse events, including sepsis, thrombocytopenia, anemia, and neutropenia [21].

Most notably, the objective response rate for patients with double-refractory disease was high, and some responses were even noted in some patients who had previously received PI3K inhibitors or an immunomodulatory treatment (such as lenalidomide), underscoring how tazemetostat can still be efficacious in heavily pre-treated patients. This may be a reflection of its novel mechanism and its first-in-class status.

Overall, tazemetostat’s unique mechanism of action, high response rate and durability of response in pre-treated patients, and relatively tolerable adverse event profile make it a promising therapy in relapsed/refractory FL in both EZH2-mutated and unmutated patients. Further investigation is needed to directly compare tazemetostat to other subsequent-line therapies and to better elucidate the predictive value of EZH2 mutational status for overall response.

### 3.7. Copanlisib in the Relapsed/Refractory Setting

Copanlisib, a PI3K inhibitor, was evaluated in the single-arm phase II CHRONOS-1 study with 104 follicular lymphoma patients who had previously progressed on two or more lines of prior therapy, including rituximab. The FL cohort demonstrated an ORR of 59% with a 20% CR rate, with a median PFS of 12.5 months and a median OS of 42.6 months at the most recent data follow-up [53,54]. Notably, 54% of patients experienced Grade 3 treatment-emergent adverse events (TEAEs), and 28.9% of patients experienced Grade 4 TEAEs, with hyperglycemia, pneumonia, neutropenia, and hypertension being the major contributors [53]. Serious infections of special interest, including pneumocystis pneumonia, pneumonitis, and colitis, were also rarely noted.

As a class, PI3K inhibitors for indolent NHLs have come under increased scrutiny by regulatory agencies, and the future of copanlisib in FL remains unclear. At the time of this writing, copanlisib monotherapy remains FDA-approved for follicular lymphoma in the third line. Future combination therapies with copanlisib also remain uncertain. In the randomized phase 3 CHRONOS-3 study where copanlisib with rituximab was compared with rituximab, overall survival appeared to be worse for the copanlisib + rituximab group during the initial 24 months of follow-up, ultimately leading to the voluntary withdrawal of the new-drug approval for copanlisib in combination with rituximab [23,64].

### 3.8. Bispecific T-Cell Engager Antibody Monotherapy in the Relapsed–Refractory Setting

Several novel bispecific T-cell engager (BiTE) antibody therapies have been studied in FL. Mosunetuzumab has been studied in the GO29781 trial, a single-arm, phase 2 study enrolling 90 patients with high-risk relapsed–refractory disease, with a high percentage of patients with POD-24 disease, high FLIPI scores, or who were refractory to the previous line of therapy [55]. Despite this high-risk population, mosunetuzumab induced a high ORR response rate of 80% with a high CR rate of 60%, a relatively high mPFS of 17.9 months, and an mOS rate that was not reached [55]. The most notable adverse events were roughly consistent with other BiTE therapies, including cytokine release syndrome (CRS), fatigue, neutropenia, immune-effector-cell-associated neurotoxicity syndrome (ICANS), and infections. Most patients with CRS had either Grade 1–2 symptoms and did not require tocilizumab or corticosteroids. Overall, the incidence of CRS or ICANS was noted to be numerically lower than in patients who had received CAR-T cell therapy while achieving high ORR and CR rates. Based on these data, mosunetuzumab has been FDA-approved in patients with relapsed–refractory FL who have progressed on two or more prior therapies [31].

Epcoritamab monotherapy has shown promise in the initial phase I/II dose-escalation study enrolling 12 R/R FL patients [56]. This study demonstrated a remarkable 90% ORR with a 50% CR rate in patients with very heavily pre-treated diseases (the median number of prior therapies was 4.5). Although this is a relatively small population, the response rate and depth of response suggest high activity of epcoritamab in R/R disease. Currently, epcoritamab monotherapy is being evaluated in 128 R/R FL patients in the larger phase I/II EPCORE NHL-1 trial in patients who have received at least two or more lines of systemic therapy. Peer-reviewed results from this larger cohort are not yet available at the time of publication.

Glofitamab monotherapy has been studied in several phase I trials—in each of these, obinutuzumab was administered as pre-treatment 7 days prior to glofitamab initiation in an effort to mitigate CRS. In a phase I open-label dose-escalation and dose-expansion study, 44 patients with Grade 1-3A FL received glofitamab [57]. ORR and CR rates were high: 70.5% and 47.7%, respectively. As with the other BiTE antibodies, major AEs included CRS, pyrexia, cytopenias, and infections. Another phase I study in the R/R setting enrolled 53 patients to receive glofitamab monotherapy [58]. The ORR was 81%, the CR rate was 70%, and the most notable and common AE was CRS with 66% incidence, the vast majority of whom experienced Grade 1 and Grade 2 events [58].

Odronextamab monotherapy has been studied in the ELM-2 phase II study, which enrolled 91 patients refractory to both anti-CD20 therapy and alkylating chemotherapy. Notably, patients here were allowed to continue odronextamab maintenance therapy after 4C of induction therapy [59]. Similar to other BiTE therapy studies, 81% of patients achieved an ORR with a 75% CR rate, and median PFS was noted to be 20.2 months. CRS, pyrexia, and cytopenias were the most notable AEs, with only one patient receiving tocilizumab for CRS management [59].

### 3.9. Bispecific T-Cell Engager Antibody Combination Therapy in the Relapsed–Refractory Setting

Building on the substantial clinical activity demonstrated in BiTE monotherapy trials, combination therapy involving BiTE therapy with other targeted therapies is currently under active investigation. Although BiTE therapy has a distinctive adverse effect profile—namely with increased incidence of cytopenias, CRS, and (less commonly) ICANS—it remains an attractive and relatively tolerable option in patients who have progressed on multiple lines of previous therapies, and combining BiTE therapy with other targeted therapies may further improve depth and duration of response.

Epcoritamab, in combination with rituximab and lenalidomide (R^2^), has been studied in early-stage trials with promising results [37,60]. In the latest update for the phase Ib/II EPCORE NHL-2 trial, a total of 109 FL patients with relapsed–refractory disease received epcoritamab + R^2^ for 12 total cycles. ORR was 97% with a complete response rate of 86%, and 6-month PFS was 93% [60]. Notably, patients with POD24 experienced numerically similar ORR and CR rates. The major adverse events were CRS (with Grade 3 or greater CRS occurring in 2% of patients) as well as ICANS in 2 patients. Although these numbers are numerically superior to R^2^ monotherapy and even epcoritamab monotherapy, this cohort of patients overall has a less-heavily pre-treated population (with median prior therapies of 1) as well as a lower percentage of patients who were refractory to prior treatment. The magnitude of additional benefit of adding R^2^ to BiTE therapy (if any), as well as the longer-term additive toxicities, therefore remains unclear. A phase III trial of epcoritamab with R^2^ in FL (EPCORE FL-1) is currently underway [38], which will give additional insights into the efficacy and long-term response of epcoritamab with R^2^.

Mosunetuzumab in combination with lenalidomide, has been investigated in a phase Ib trial enrolling 27 patients with R/R FL [61]. These patients received mosunetuzumab step-up therapy during the first cycle, followed by lenalidomide therapy in subsequent cycles. ORR was 92%, and complete response was documented in 77% of patients, supporting the high activity of this combination. CRS was the most common adverse event and was limited to Grade 1 or Grade 2, with no patients requiring tocilizumab, ICU admission, vasoactive agent support, or high-flow oxygen support. Grade 3–4 neutropenia was reported in 19% of patients. As with the epcoritamab with R^2^ combination, it is currently unclear whether there is significant additive benefit of mosunetuzumab with lenalidomide when compared to mosunetuzumab monotherapy, and further studies are underway to further assess this combination. The CELESTIMO phase III trial comparing mosunetuzumab with lenalidomide with R^2^ is currently recruiting patients [65].

Glofitamab, in combination with obinutuzumab, has been studied in a non-randomized phase I trial by Morschhauser et al. alongside three cohorts who received glofitamab monotherapy [58]. The 19 patients enrolled in the combination cohort received obinutuzumab concurrently with glofitamab in addition to the obinutuzumab received prior to glofitamab initiation. The ORR for the combination cohort of 100% and CR rate of 73.7% compared favorably to the ORR of 81% in the monotherapy cohort and the 70% CR rate in the monotherapy cohort. CRS rates were numerically higher in the combination cohort vs. the monotherapy cohort (78.9% vs. 66%), and neutropenia was also more prevalent in the combination cohort (58% vs. 26%). Although this study involves a small number of patients, and patients were not randomized to the treatment groups, it appears that the addition of obinutuzumab may have higher response rates at the cost of increased AEs, including CRS and cytopenias. As is the case with other BiTE/targeted therapy combinations, further randomized comparator studies are needed to better assess the potential benefit and potential additional adverse events when compared to BiTE monotherapy.

Overall, BiTE therapy—either as monotherapy or in combination with other agents—shows significant promise in the R/R setting for patients who have progressed on multiple lines of previous therapy and results in high response rates as well as deep and often durable responses. This high degree of clinical activity must be balanced against the relatively higher rates of serious adverse events when compared to other targeted therapies, especially those related to CRS, ICANS, cytopenias, and infections. It is less clear currently if the additional toxicity of combination therapies involving BiTE with targeted therapy will translate to substantial long-term clinical benefit (i.e., PFS or OS). Also unclear is the proper sequencing of BiTE therapy with regard to targeted therapies and chemoimmunotherapy. Further studies are needed to directly compare BiTE therapy to cellular therapies or other targeted therapies, as well as to explore further combinations of BiTE therapy with targeted therapies.

### 3.10. Chimeric Antigen Receptor T-Cell Therapy in the Relapsed–Refractory Setting

CAR-T therapy has been studied in FL in the relapsed–refractory setting. Axicabtagene ciloleucel (axi-cel) was studied in the single-arm, phase II ZUMA-5 trial, which enrolled 124 patients with FL [43]. Overall, these patients were high-risk, with 44% with high-risk (≥3) FLIPI scores, a median of three prior lines of therapy, 68% who were previously refractory to the last therapy, and 55% with POD24 disease [40]. High response rates were noted, with a 94% ORR rate and a 79% CR rate. At 18 months, PFS and OS were 64.8% and 87.4%, respectively [43]. CRS occurred in 78% of patients, neurological events occurred in 56% of patients, and Grade 3 or worse adverse events occurred in 85% of patients, with cytopenias (70%) and infections (18%) being the most frequent [43]. Notably, one patient’s death from multiorgan system failure was directly attributed to axi-cel treatment. Based on these data, axi-cel has received FDA approval in the U.S. for relapsed refractory FL after two or more lines of therapy. Axi-cel represents a highly effective option even for patients with highly refractory, heavily pre-treated, or otherwise high-risk disease, although the tradeoffs include high rates of serious adverse events, including CRS, ICANS, infections, and cytopenias. Future comparative studies are warranted to assess axi-cel against other therapies, most notably BiTE therapy.

Tisagenlecleucel was studied in the phase II ELARA study, which enrolled 97 patients with R/R FL [42]. Compared with the ZUMA-5 study, the ELARA study enrolled a higher-risk population: higher percentages of patients with high-risk FIPI scores, patients who were refractory to the last therapy, and patients with POD24 disease. As in the ZUMA-5 trial, a high ORR rate (86.2%) and CR (69.1%) rate were noted [42]. CRS occurred in 49% of patients, all of whom were Grade 1–2, in contrast to the 6% of pts in the ZUMA-5 study who had Grade 3 or greater CRS. Neurological events occurred in 37.1% and ICANS developed in 4.1% of patients, and infections occurred in 18.6% of patients [42]. This adverse event profile appears more favorable than the ZUMA-5 patients, although no direct comparisons yet exist.

Data from a preliminary analysis of the TRANSCEND FL study, which enrolled 130 patients with R/R FL, show that lisocabtagene maraleucel (liso-cel) treatment results in a high ORR (97.0%) and CR rate (94.1%) [44]. CRS occurred in 58% of patients, with 1% of patients experiencing Grade 3 or higher CRS. As with the other CAR-T-based therapies, cytopenias and infections were among the prominent adverse events. In this study, one patient died from Grade 5 macrophage activation syndrome. Although data have yet to mature, this initial analysis shows promising results for liso-cel.

Overall, CAR-T therapy shows promise in follicular lymphoma, with high response rates and deep and durable responses. However, the rates of adverse events are generally higher than in BiTE therapy, which also produces high response rates in patients with high-risk diseases. This includes the hallmark CRS and ICANS of cytotoxic T-cell therapy, as well as higher rates of cytopenias and infections, likely related to initial lymphodepleting regimens that are required for CAR-T therapy. Furthermore, CAR-T requires initial leukapheresis and weeks of manufacturing lead time, while BiTE therapy is available “off the shelf.” It currently remains unclear if the potentially higher response rates of CAR-T are worth these distinct disadvantages. Further comparison studies are needed, especially in light of novel emerging BiTE/targeted therapy combinations.

## 4. Therapies in the Front-Line Setting

Chemotherapy in combination with rituximab or obinutuzumab, followed by rituximab maintenance therapy, has long remained the front-line treatment-of-choice for follicular lymphoma, with long-term follow-up showing median PFS on the order of 10 years [6,8]. In this context, targeted therapies in the front-line setting must clear a relatively high efficacy standard. However, targeted therapy approaches have distinct advantages in terms of cumulative adverse events and tolerability when compared to chemotherapy. In this context, recent advances in targeted therapy for follicular lymphoma have extended to the front-line setting. Table 2 summarizes the relevant studies supporting the use of targeted therapies in follicular lymphoma in the front-line setting. The most robust data for targeted therapy in this space revolve around combination lenalidomide + anti-CD20 therapy, although new approaches involving BTK inhibition with other targeted therapies also show promise.

### 4.1. Rituximab with Lenalidomide in the Front-Line Setting

The combination of rituximab with lenalidomide (R^2^) was evaluated in the first-line setting in the phase III randomized RELEVANCE trial, which directly compared rituximab/lenalidomide to rituximab with investigator-choice chemotherapy. A total of 1030 patients with previously untreated FL who were in need of treatment by GELF criteria were randomly assigned 1:1 to R^2^ or rituximab plus chemotherapy. For the patients in the rituximab plus chemotherapy group, 72% received cyclophosphamide, doxorubicin, vincristine, and prednisone (CHOP); 23% received bendamustine; and 5% received cyclophosphamide, vincristine, and prednisone (CVP).

In the most recent six-year update on RELEVANCE, R^2^ showed no significant difference in overall survival (six-year OS 89% in both groups) or progression-free survival (six-year PFS 60% in rituximab/lenalidomide vs. 59% in rituximab/chemotherapy) [66]. Notably, similar rates of histologic transformation to aggressive lymphoma, overall response rate to subsequent treatment, progression of disease within 24 months (POD24), and five-year overall survival in patients with POD24 were seen in both groups. Although the overall rates of Grade 3 or Grade 4 adverse events were similar in the two groups, the adverse-effect profile of the R^2^ group differed from the rituximab/chemotherapy group. Lower incidences of cytopenias, fatigue, nausea, vomiting, and peripheral neuropathy were seen in the R^2^ group, but higher incidences of adverse effects traditionally associated with lenalidomide were seen: diarrhea, rash, and cutaneous reactions [67]. As many adverse effects of traditional chemotherapy are cumulative and ultimately dose-limiting, R^2^ provides a reasonable front-line alternative for patients who may not tolerate the toxicities of chemotherapy well.

Notably, a subsequent analysis of the French Lymphoma Study Association (LYSA) RELEVANCE subgroup demonstrated that complete molecular responses (as measured by a PCR assay of the BCL2-JH translocation) were seen more commonly with the R^2^ arm, which translated to a significantly increased PFS when compared to those patients who had positive BCL2-JH by PCR at the end of induction therapy [68]. Although this ultimately did not translate into a PFS benefit, this suggests that R^2^ may be more effective at inducing deeper initial responses to treatment.

Although the RELEVANCE study ultimately failed to demonstrate the superior efficacy of R^2^ when compared to rituximab/chemotherapy, the similarity of outcomes between the two treatments suggests that R^2^ is a reasonable alternative with an adverse effect profile that may be better tolerated in the long term. R^2^ is, therefore, an appealing option for frail patients or for patients in whom cumulative chemotherapy-related toxicities are significantly burdensome.
cancers-15-04483-t002_Table 2Table 2Summary of studies supporting the use of targeted therapies in the front line for follicular lymphoma. (FL: follicular lymphoma; FLIPI: Follicular Lymphoma International Prognostic Index; ORR: overall response rate; CR: complete response rate; PFS: progression-free survival; OS: overall survival; POD24: progression of disease after 24 months of initial therapy; AEs: adverse events; NR: not reached; mo: months; yr.: years; N/A: not available).Referenced StudyPhaseTherapies StudiedFL GradesFLIPI ScoreORR (CR)PFS (Median)OS (Median)POD24 (%)Notable AEs [66,67] (RELEVANCE)IIIRituximab + Lenalidomide1–3a0–1: 15%2: 36%3–5: 49%61%(48%)3 yr: 77%6 yr: 60%(NR)3 yr: 94%6 yr: 89%(NR)13%Cutaneous reactions, diarrhea, rash, neutropenia [69](GALEN)Ib/IIObinutuzumab + Lenalidomide1–3a0–1: 17%2: 40%3–5: 43%92%(47%)3 yr: 82%(NR)3 yr: 94%(NR)14%Asthenia, neutropenia, constipation, diarrhea, cough [13](PCYC-1125-CA)IIIbrutinib + Rituximab1–3a0–1: 12%2: 38%3–5: 50%85% (40%)30 mo: 67%(41.9 mo)30 mo: 97%(NR)N/AFatigue, diarrhea, nausea, bleeding, cardiac events [14]IIObinutuzumab + Ibrutinib + Venetoclax1–3a0–1: 12.5%2: 25%3–5: 62.5%100% (100%)12 mo: 100% (NR)12 mo: 100% (NR)N/AFatigue, lymphopenia, diarrhea, neutropenia, rash, thrombocytopenia

### 4.2. Obinutuzumab with Lenalidomide in the Front-Line Setting

Initial trials of obinutuzumab with lenalidomide (the GALEN regimen) have demonstrated meaningful clinical benefit in the front-line setting. A combined, non-randomized phase Ib/II study of the GALEN regimen in the front-line setting for 100 patients with treatment-naïve FL showed a 92% ORR at the end of induction, including a 47% complete response rate [69]. Notably, the complete response rate at 30 months (CR30)—a surrogate marker for PFS—was 63% in this study, compared with 48% in the R^2^ arm of RELEVANCE [69]. The 3-year PFS of 82% and 3-year OS of 94% are numerically similar to the R^2^ arm in RELEVANCE (3-year PFS 77% and 3-year OS 94%), although comparisons across trials should be interpreted with caution, and these regimens have not yet been compared head-to-head. The POD24 rate of patients who received GALEN was 14%, numerically similar to the POD24 rate of patients receiving R^2^ in the RELEVANCE study.

The overall adverse event profile was similar numerically to the patients in the GALEN study in the refractory/relapsed setting, with neutropenia and especially Grade 3 or higher neutropenia being the most prominent. Notably, 75% of patients experienced an infection during treatment, with respiratory tract infections as the most common manifestation, which highlights the need for clinical vigilance for infectious complications in patients treated with GALEN.

Although large, randomized clinical trials comparing the GALEN regimen against chemotherapy and against R^2^ are needed to further assess the potentially superior PFS and CRR rate of GALEN, these initial results show promise and provide another chemotherapy-free alternative in the front-line setting. Further unanswered questions include the duration of treatment—in this trial, patients received initial induction of obinutuzumab and lenalidomide, followed by one year of maintenance lenalidomide with obinutuzumab, followed by one subsequent year of obinutuzumab monotherapy. It is currently unclear whether a shorter duration of maintenance therapy would produce similar outcomes and perhaps fewer adverse events.

### 4.3. Ibrutinib in Combination with Anti-CD20 Therapy and Venetoclax in the Front-Line Setting

Attempts to improve on the relatively disappointing performance of ibrutinib monotherapy in the relapsed–refractory setting have resulted in combinations of ibrutinib with the anti-CD20 agents rituximab and obinutuzumab. More recently, a pair of phase II studies have studied these combinations in the front-line setting.

The PCYC-1125-CA trial enrolled a total of 80 patients in two arms: Arm 1 with both ibrutinib and rituximab started concurrently and Arm 2 with a two-month ibrutinib lead-in period, followed by concurrent ibrutinib/rituximab therapy. In Arm 1 (the main study arm), the ORR was 85% with a 40% CR rate [13]. These figures compare favorably to single-agent rituximab and are numerically comparable to the rituximab–chemotherapy arm of the RELEVANCE trial. The 30-month PFS rate of 67% is numerically inferior to R^2^ in RELEVANCE and the GALEN regimen in the front-line setting, although such comparisons should be made with caution given the differences between the three studies [13,67,69]. The most prominent AEs were fatigue, diarrhea, nausea, and myalgia—and were overall similar to the AE profile of ibrutinib monotherapy in other clinical settings. Of particular interest is the rate of bleeding and cardiac events for which ibrutinib has been previously implicated. In this study, bleeding was present in 40% of patients, although only 2.5% of patients experienced Grade 3 or 4 bleeding events. Cardiac AEs occurred in 14% of patients [13].

Of particular note is the limited duration of rituximab to four weekly infusions over one month—this is dramatically different from the two years of induction and maintenance anti-CD20 therapy used in the R^2^ and GALEN regimens. The relatively short duration of rituximab therapy in PCYC-1125-CA is likely a factor in the similar AE profile of rituximab/ibrutinib when compared to ibrutinib monotherapy. It is unclear whether this fixed-duration of anti-CD20 therapy could account for the differences in PFS and OS when compared to R^2^ and GALEN, whether a longer duration of rituximab with ibrutinib could improve duration and depth of responses, and whether such a longer duration of rituximab with ibrutinib would result in greater toxicities. The ongoing phase 3 PCYC-1141-CA PERSPECTIVE study is further evaluating ibrutinib/rituximab against rituximab monotherapy in unfit and elderly patients, which may shed more light on these questions [13].

Ongoing studies in this space also include a phase II trial of a combination of obinutuzumab with ibrutinib and venetoclax. An initial report of the first eight patients enrolled showed a 100% complete response rate, PFS rate, and OS rate at a median time to follow-up of 12.2 months [14]. Adverse events included fatigue, cytopenias, diarrhea, and rash. While these represent only initial results and are limited by small numbers and limited duration of follow-up, they suggest that combining ibrutinib with venetoclax and rituximab could offer substantial efficacy in the front line for follicular lymphoma.

Overall, ibrutinib, in combination with other agents (rituximab with or without venetoclax), shows promise as a tolerable first-line therapy alternative to chemotherapy-based regimens. More studies are needed to better characterize the optimal combination of these therapies to maximize efficacy while limiting toxicity and to compare ibrutinib-based regimens to R^2^ and GALEN.

## 5. The Role of Molecular Testing in Selecting Targeted Therapies

Although current treatment paradigms do not require molecular testing to select therapy in FL, molecular markers may become useful in the future to help predict response. This would be particularly helpful in selecting therapies in the relapsed–refractory setting, where the presence or absence of certain biomarkers may be informative in selecting subsequent-line treatments.

In a post-hoc analysis of the GALLIUM study, patients with EZH2 mutations have been shown to have an increased PFS with CHOP/CVP-based regimens than those who were EZH2 wild type [70]. Notably, the same trend was not demonstrated with bendamustine-based treatment [70]. This suggests that EZH2 mutational status is a differential predictor of response to certain chemotherapies.

However, in the phase II trial evaluating tazemetostat, the predictive role of EZH2 mutational status remains unclear. The apparent numerically superior ORR of those patients with EZH2 mutation status seems to underscore EZH2′s importance as a driver mutation for FL and suggests that EZH2-mutant disease is more vulnerable to EZH2 inhibition [21]. As a result, the U.S. FDA has approved a companion diagnostic EZH2 mutation test for tazemetostat. However, a subsequent analysis suggests that clinical differences in the groups could have accounted for these differences—notably, that the EZH2 wild-type group overall had poorer prognostic factors than the EZH2 mutant group. After a propensity-matched analysis, the corrected ORR and median PFS were not statistically different between the two groups [71]. This suggests that while EZH2 mutational status may still be a predictor of response to tazemetostat, even EZH2 wild-type FL is dependent on EZH2 signaling to some extent.

In the study by Bartlett et al. evaluating ibrutinib in the relapsed–refractory setting, 31/40 patients also underwent genetic sequencing of their tumors, and patients with CARD11 mutations had significantly inferior response rates than those who were CARD11 wild-type [11]. Additionally, IGLL5 mutations were associated with improved PFS, and KMT2D-mutant and FOXO1-mutant patients had a longer duration of response [11]. Although the roles of CARD-11, IGLL5, KMT2D, and FOXO1 mutational status in predicting response to BTK inhibitor therapy has not yet been established in follicular lymphoma, these findings may prove useful in the future when selecting from a host of targeted therapies.

The CONTRALTO study attempted to correlate responses to venetoclax therapy with BCL-2 IHC and FISH results, but PFS was not noted to be different amongst BCL-XL high vs. low staining, nor did expression of BCL-2 family genes correlate with differential response rates [52].

Initial studies with lenalidomide in patients with relapsed/refractory NHL showed that a subset of responders achieved durable and long-lasting responses, suggesting a differential response that may be underpinned by differing molecular characteristics [72]. Although no molecular markers predictive of response to R^2^ in FL have yet been identified, the absence of NOTCH pathway mutations in CLL has been shown to predict better response to lenalidomide/rituximab therapy [73]. Notably, lenalidomide, in combination with rituximab (R^2^), also seems to negate the predictive value of FCGR3A polymorphisms in predicting response to rituximab-based therapy [74]. In a phase II study of R^2^ in heavily pre-treated NHL patients, 22/27 patients were assessed for FCRG3A polymorphisms, and 21/22 patients had the low-affinity polymorphisms (V/F or F/F) that would typically predict poor responses to rituximab-based therapy. Despite this, the ORR for patients with these low-affinity phenotypes still demonstrated numerically high ORR of 73% (V/F) and 70% (F/F), which were not significantly different [74]. This suggests that, in patients with relapsed–refractory NHL, R^2^ may be superior in terms of ORR when compared to rituximab–chemotherapy combinations.

In the CHRONOS-1 study, gene expression profiling was performed in 54 patients with FL. Patients with upregulation of gene sets related to PI3K/BCR pathway signaling showed overall higher response rates to copanlisib [54]. Additionally, patients with lower macrophage gene expression profiles (as determined by weighted gene expression scores) were found to have a trend towards improved response [54]. Overall, these findings suggest that molecular profiling may be helpful in predicting potential response to PI3K inhibitors such as copanlisib and that upregulation of genes in the PI3K/BCR pathway or a lower macrophage gene expression profile score may be predictive of response.

Overall, more investigation into biomarkers as predictors of response to targeted therapy is warranted and could be helpful in determining optimal targeted therapy regimens in both the front-line and the relapsed/refractory setting.

## 6. Conclusions and Future Directions

In the last few years, multiple targeted therapies have emerged to treat FL in both the front-line and relapsed/refractory setting. While the most robust phase III randomized clinical trial (RCT) evidence exists only for R^2^ in both treatment settings, several novel therapeutic agents, including BTK inhibitors, BCL-2 inhibitors, EZH-2 inhibitors, and PI3K inhibitors, also demonstrate promise in the treatment of FL. Several RCTs are currently underway to evaluate these therapies.

Novel combinations of these targeted therapies are currently being studied to maximize efficacy and optimize adverse effect profiles. This includes adding targeted therapy to the R^2^ backbone, such as the phase 1b/3 SYMPHONY-1 trial (tazemetostat with R^2^). Future studies may also use the GALEN regimen as a backbone. Perhaps the most exciting combinations involve BiTE therapy and targeted therapy due to high efficacy, and further investigation is underway for novel combinations of BiTE therapy with targeted therapy in both the relapsed–refractory setting as well as the front-line setting [65,75].

Future directions also include larger-scale RCTs for some of these regimens that show promise. Direct comparison amongst these regimens may be helpful in the optimal management of certain subsets of patients, such as those with POD24, rituximab-refractory, or chemotherapy-refractory disease. Further investigation of molecular predictors of response may also prove helpful for individualizing the treatment regimen of choice, especially with multiple potential regimens that all show promise.

Targeted therapies offer better-tolerated and, in some cases, more effective treatment than traditional rituximab–chemotherapy approaches. Multiple targeted options are now available to patients who are refractory to rituximab/chemotherapy or who cannot tolerate chemotherapy due to adverse effects. Adverse effect profiles for targeted therapies are distinct, and many AEs of targeted therapies are more easily managed and more tolerable in the long term when compared to the cumulative toxicities (such as fatigue, myelosuppression, and neuropathy) of chemotherapy. Additionally, deeper remissions with targeted therapy can sometimes be achieved, which may herald longer PFS and OS in these patients. Overall, targeted therapies offer a new paradigm for treating FL that allows providers and patients to move beyond traditional chemoimmunotherapy.

## Figures and Tables

**Figure 1 cancers-15-04483-f001:**
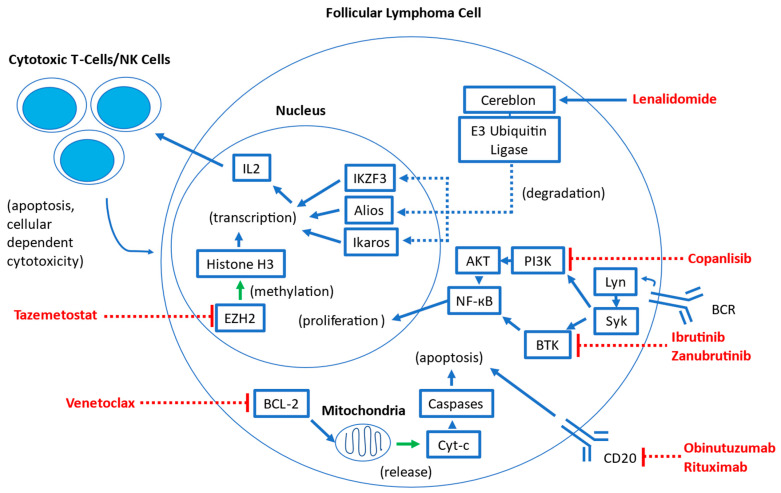
Overview of mechanisms of targeted therapies in follicular lymphoma. Abbreviations: BCR: B-cell antigen receptor; Lyn: LYN Lck/Yes novel tyrosine kinase; Syk: spleen tyrosine kinase; BTK: Bruton tyrosine kinase; PI3K: phosphoinositide 3-kinase; AKT: Protein Kinase B; NK-кB: Nuclear factor kappa B; BCL-2: B-cell Lymphoma 2; Cyt-c: Cytochrome C; EZH2: enhancer of zeste homolog 2; IKZF3: Ikaros family zinc finger 3; IL2: Interleukin-2; NK: Natural Killer.

**Figure 2 cancers-15-04483-f002:**
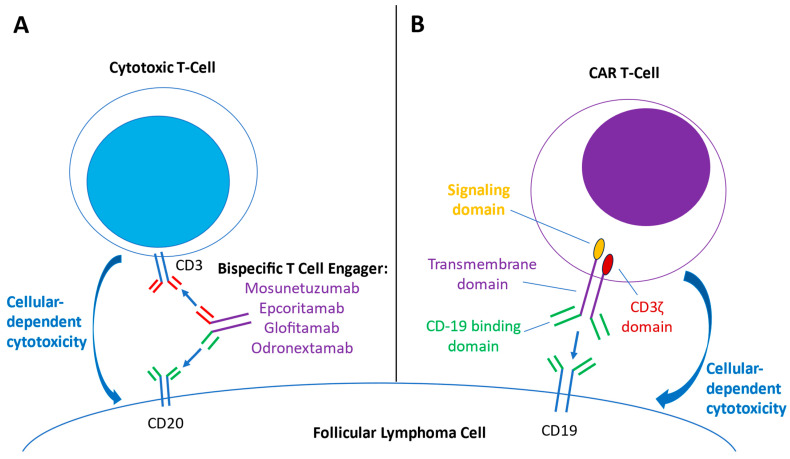
Overview of mechanisms of bispecific T-cell engager (BiTE) therapy and Chimeric antigen receptor (CAR) T-cell therapy. (**A**) BiTE antibodies contain both anti-CD3 binding domains (red) and anti-CD-20 binding domains (green). The engagement of both anti-CD3 and CD20 domains brings cytotoxic T cells in close proximity to the follicular lymphoma cell, creating an immune synapse and activating cytokine transcription and other pathways that result in antitumor cellular-dependent cytotoxic effects. (**B**) Engineered CAR-T cells express chimeric antigen receptors, consisting of a CD-19 binding domain (green), a transmembrane domain (purple), a costimulatory signaling domain (yellow), and a CD3 ζ domain (red). As the CAR-T cell binds CD19 on the lymphoma cell, the signaling and CD3 ζ domains are activated, activating downstream signaling and transcription factors that ultimately lead to tumor-directed cytotoxic effects.

**Table 1 cancers-15-04483-t001:** Summary of studies supporting the use of targeted therapies in the relapsed–refractory setting for follicular lymphoma. (FL: follicular lymphoma; POD24: progression of disease after 24 months of initial therapy; FLIPI: Follicular Lymphoma International Prognostic Index; ORR: overall response rate; CR: complete response rate; PFS: progression-free survival; OS: overall survival; AEs: adverse events; NR: not reached; mo: months; yr: years; N/A: not available; EZH2mt: mutation in EZH2; EZH2wt: non-mutated EZH2; CRS: cytokine release syndrome; ICANS: immune-effector-cell-associated neurotoxicity syndrome).

Referenced Study	Phase	Agents Studied	FL Grades	Percentage POD24 pts	FLIPI Score Distribution	Median Prior Therapies	Refractory to Last Therapy	ORR (CR)	PFS (Median)	OS (Median)	Notable AEs
[48,49] (AUGMENT)	III	Lenalidomide + Rituximab	1–3a	31%	0–1: 29% 2: 31% 3–5: 39%	1	17%	78% (34%)	2 yr: 58% (39.4 mo)	2 yr: 93% (NR)	Pulmonary embolism, infection, neutropenia, cutaneous reactions
[50] (GALEN)	II	Obinutuzumab + Lenalidomide	1–3a	27%	0–1: 23% 2: 35% 3–5: 42%	2	17%	79% (38%)	2 yr: 65% (NR)	2 yr: 87% (NR)	Asthenia, cytopenias, rash, bronchitis, diarrhea
[51]	I/II	Obinutuzumab + Lenalidomide	1–3a	N/A	N/A	2	N/A	98% (72%)	2 yr: 73%	N/A	Neutropenia, thrombocytopenia, fatigue, rash, cough
[52] (CONTRALTO)	II	Venetoclax + Rituximab + Bendamustine	1–3a	N/A	N/A	3	37.3%	84% (75%)	18 mo: 61.7%	N/A	Cytopenias, nausea, vomiting, diarrhea, fatigue
		Venetoclax + Rituximab	1–3a	N/A	N/A	3	50.0%	35% (17%)	18 mo: 26.9%	N/A	Cytopenias, diarrhea, infusion reactions
[11]	II	Ibrutinib	1–3a	N/A	0–1: 15% 2: 35% 3–5: 50%	3	35%	38% (13%)	2 yr: 20.4% (14.0 mo)	2 yr: 79% (NR)	Neutropenia, anemia, infection, diarrhea
[15] (ROSEWOOD)	II	Zanubrutinib + Obinutuzumab	N/A	28%	0–1: N/A 2: N/A 3–5: 53%	3	32%	68.3% (37.2%)	(27.4 mo)	18 mo: 85.4% (NR)	Thrombocytopenia, neutropenia, diarrhea, fatigue, constipation
[21]	II	Tazemetostat	1–3b	42% (EZH2mt)	N/A	2 (EZH2mt)	49% (EZH2mt)	69% (13%) (EZH2mt)	(13.8 mo) (EZH2mt)	(NR) (EZH2mt)	Sepsis, anemia, neutropenia, thrombocytopenia
				59% (EZH2wt)	N/A	3 (EZH2wt)	41% (EZH2wt)	35% (4%) (EZH2wt)	(11.1 mo) (EZH2wt)	(NR) (EZH2wt)	
[53,54] (CHRONOS-1)	II	Copanlisib	1–3a	N/A	N/A	3	61%	59% (20%)	2 yr: 34% (12.5 mo)	2 yr: 69% (42.6 mo)	Serious infections, pneumocystis, hyperglycemia, hypertension, neutropenia
[55] (GO29781)	II	Mosunetuzumab	1–3a	52%	0–1: 29% 2: 27% 3–5: 44%	3	69%	80.0% (60.0%)	18 mo: 47.0% (17.9 mo)	18 mo: NR (NR)	CRS, neutropenia, tumor flare reaction, infections, ICANS
[56]	I/II	Epcoritamab	N/A	N/A	N/A	4.5	83%	90% (50%)	N/A	N/A	CRS, pyrexia, fatigue, anemia, dyspnea
[57]	I	Glofitamab	1–3a	N/A	N/A	N/A	N/A	70.5% (47.7%)	(11.8 mo)	N/A	CRS, neutropenia, pyrexia, thrombocytopenia, anemia, fatigue
[58]	I	Glofitamab	1–3a	36%	0–1: N/A 2: N/A 3–5: 53%	3	53%	81% (70%)	N/A	N/A	CRS, neurologic AE, pyrexia, neutropenia
[59] (ELM-2)	II	Odronextamab	1–3a	48%	0–1: N/A 2: N/A 3–5: 58%	3	74%	81% (75%)	(20.2 mo)	(NR)	CRS, pyrexia, anemia, infusion reactions
[60] (EPCORE NHL-2)	Ib/II	Epcoritamab + Lenalidomide + Rituximab	N/A	38%	0–1: N/A 2: N/A 3–5: 56%	1	49%	97% (86%)	6 mo: 93%	N/A	CRS, neutropenia, fatigue, ICANS
[61]	Ib	Mosunetuzumab + Lenalidomide	1–3a	11%	N/A	1	N/A	92% (77%)	N/A	N/A	CRS, neutropenia
[58]	I	Glofitamab + Obinutuzumab	1–3a	53%	0–1: N/A 2: N/A 3–5: 58%	2	42%	100% (73.7%)	N/A	N/A	CRS, neurologic AE, pyrexia, neutropenia, thrombocytopenia
[43] (ZUMA-5)	II	Axicabtagene ciloleucel	1–3a	55%	0–1: 18% 2: 39% 3–5: 44%	3	68%	94% (79%)	18 mo: 64.8% (NR)	18 mo: 87.4% (NR)	CRS, ICANS, hypotension, cytopenias, infections
[42] (ELARA)	II	Tisagenlecleucel	1–3a	62.9%	0–1: NA 2: NA 3–5: 59.8%	4	78.4%	86.2% (69.1%)	12 mo: 67% (NR)	(NR)	CRS, ICANS, hypotension, cytopenias, infections
[44] (TRANSCEND FL)	II	Lisocabtagene Maraleucel	N/A	43%	0–1: NA 2: NA 3–5: 57%	3	N/A	97.0% (94.1%)	12 mo: 80.7% (NR)	N/A	CRS, neurologic AE, cytopenias, infections

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
