# Peer review of "Targeted Therapy in Follicular Lymphoma: Towards a Chemotherapy-Free Approach"

_cancers, 2023, doi:10.3390/cancers15184483_

Round 1

Reviewer 1 Report

The work summarizes well the results of the new targeted therapies in follicular lymphoma. However, there is a serious deficiency in the complete absence of any hints of bispecific antibody or CAR t data in follicular lymphoma.

In fact, in my opinion, bispecifics currently have the most important and promising data in follicular lymphoma both in single agents and in combination. Epcoritamab, GLofitamab and Mosunetuzumab, alone or in combination with Lenalidomide, cannot fail to be part of a review on target therapies where, on the other hand, too much space is given, for example, to data on Venetoclax or Copanlisib, now molecules without a particular therapeutic future in the follicular. I don't expect it to turn into a review of CAR t, but don't even mention them, it seems out of time. In addition, other very interesting PI3K inhibitors are not mentioned which, due to problems more of drug economy than of results, have suddenly ceased development but of which the first data presented were already known, such as Parsaclisib or Zandelisib.

I believe that the work should be updated with the aforementioned topics but, in particular, it is unacceptable that a part on bispecifics or CAR T is absent.

Author Response

We greatly appreciate the helpful recommendations submitted by the reviewer. Point 1: However, there is a serious deficiency in the complete absence of any hints of bispecific antibody or CAR-T data in follicular lymphoma…Epcoritamab, Glofitamab, and Mosunetuzumab, alone or in combination with lenalidomide, cannot fail to be part of a review on target therapies where, on the other hand, too much space is given, for example to data on Venetoclax or Copanlisib, now molecules without a particular therapeutic future in the follicular. Response 1: We have added sections 3.8, 3.9, and 3.10 addressing BiTE therapy, BiTE therapy combinations, and CAR-T in the relapsed refractory setting, respectively. We have also added corresponding paragraphs introducing CAR-T and BiTE therapy in section 2. We have added summaries of the relevant BiTE, BiTE in combination with targeted therapy, and CAR-T data to Table 1. We have updated the introduction and conclusion areas and the abstract to include BiTE and CAR-T. We have added Figure 2, which illustrates the mechanisms for BiTE therapy and CAR-T. Point 2: In addition, other very interesting PI3K inhibitors are not mentioned, which, due to the problems more of drug economy than results, have suddenly ceased development but of which the first data presented were already known, such as parsaclisib or Zandelisib. Response 2: Have added a discussion about parsaclisib and zandelisib, including initial data regarding their efficacy and the circumstances surrounding the cessation of development for both therapies.

Reviewer 2 Report

As the manuscript team is one out of many global teams working on follicular lymphoma therapies, they encourage to add answers for how they managed the literature survey on the FL treatment. it easy to recognize numerous published articles dealing with the same specific point. Please add where you search; PubMed, google scholar, web of Science, Scopus... etc. to cover your issue, to be sure that you covered all published articles on Targeted Therapy in Follicular Lymphoma using a single, combination of mAbs with other curable drugs.

The figure offers a view of the single/combination therapies, but it needs to be more molecular detail.

Knowing that this does not reduce the effort and value of the manuscript in its current form, and the required additions are to increase the value of the manuscript.

Author Response

We greatly appreciate the helpful recommendations submitted by the reviewer.

Point 1: As the manuscript team is one out of many global teams working on follicular lymphoma therapies, they are encouraged to add answers for how they managed the literature survey on the FL treatment. It easy to recognize numerous published articles dealing with the same specific point. Please add where you searched: PubMed, Google Scholar, Web of Science, Scopus, etc. to cover your issue, to be sure that you covered all published articles on Targeted Therapy in Follicular Lymphoma using a signle, combination of mAbs with other curable drugs.

Response 1: We have added in the introduction that PubMed was used to search for articles and that we have identified the sentinel articles related to targeted treatment for follicular lymphoma, in keeping with the narrative review format of the manuscript.

Point 2: The figure offers a view of the single/combination therapies, but it needs to be more molecular detail.

Response 2: We have revised the Figure 1 to show more molecular detail for the venetoclax, tazemetostat, lenalidomide, and BCR/BTK/PI3K pathways. We have also added Figure 2 as an overview of BiTE therapy and CAR-T therapy.

Reviewer 3 Report

Chen et al performed a comprehensive narrative review on the recent innovations in the first-line therapy and in the therapy for relapsed/refractory Fl. 

The review is in line with current therapeutic approaches and aerea of investigations for FL and can be an useful tool for a rapid update of the reader .  

Author Response

We thank the reviewer for his or her helpful comments. 

Round 2

Reviewer 1 Report

I think there has been a major revision and in-depth study of the paper which has now taken into consideration all the most recent therapeutic innovations and all this has radically changed its structure, figures and tables.

Now I think it's an excellent review, maybe it needs one last clarification on the meaning of the term "target thrapy" (are Bites and CAR Ts not targets?) but it certainly deserves publication

I think there has been a major revision and in-depth study of the paper which has now taken into consideration all the most recent therapeutic innovations and all this has radically changed its structure, figures and tables.

Now I think it's an excellent review, maybe it needs one last clarification on the meaning of the term "target thrapy" (are Bites and CAR Ts not targets?) but it certainly deserves publication

Reviewer 2 Report

Thank you for the revision!